# Transgenic HA-1-Specific CD8^+^ T-Lymphocytes Selectively Target Leukemic Cells

**DOI:** 10.3390/cancers15051592

**Published:** 2023-03-03

**Authors:** Artem Pilunov, Dmitrii S. Romaniuk, Anton Shmelev, Savely Sheetikov, Anna N. Gabashvili, Alexandra Khmelevskaya, Dmitry Dianov, Ksenia Zornikova, Naina T. Shakirova, Murad Vagida, Apollinariya Bogolyubova, Grigory A. Efimov

**Affiliations:** National Hematology Research Center, 125167 Moscow, Russia

**Keywords:** adoptive transfer, acute myeloid leukemia, transgenic TCR, allo-HSCT, minor histocompatibility antigens, HA-1

## Abstract

**Simple Summary:**

A relapse of the malignant disease frequently occurs after allogeneic hematopoietic stem cell transplantation. Immune recognition of minor histocompatibility antigens, the polymorphic peptides that differ between donor and recipient, often triggers a beneficial graft-versus-leukemia response. The transgenic donor-derived cytotoxic T cells, which recognize patient-specific minor histocompatibility antigens presented by hematopoietic cells, allow precise elimination of malignant recipient cells while sparing both donor and non-hematopoietic patient cells. We generated the MiHA-specific T cells by gene editing to knock out the endogenous T cell receptor, followed by lentiviral transduction of HA-1-specific T cell receptors. Modified T cells demonstrated cytotoxicity against leukemia cells from HA-1+ donors with acute myeloid leukemia, acute T-cell, and B-cell lymphoblastic leukemia. Transgenic T cells showed no cytotoxicity against donor cells lacking HA-1 surface presentation. The proposed therapeutic approach could be used after allogeneic hematopoietic stem cell transplantation to prevent and treat leukemia relapse.

**Abstract:**

A significant share of allogeneic hematopoietic stem cell transplantations (allo-HSCT) results in the relapse of malignant disease. The T cell immune response to minor histocompatibility antigens (MiHAs) promotes a favorable graft-versus-leukemia response. The immunogenic MiHA HA-1 is a promising target for leukemia immunotherapy, as it is predominantly expressed in hematopoietic tissues and presented by the common HLA A*02:01 allele. Adoptive transfer of HA-1-specific modified CD8^+^ T cells could complement allo-HSCT from HA-1- donors to HA-1+ recipients. Using bioinformatic analysis and a reporter T cell line, we discovered 13 T cell receptors (TCRs) specific for HA-1. Their affinities were measured by the response of the TCR-transduced reporter cell lines to HA-1+ cells. The studied TCRs showed no cross-reactivity to the panel of donor peripheral mononuclear blood cells with 28 common HLA alleles. CD8^+^ T cells after endogenous TCR knock out and introduction of transgenic HA-1-specific TCR were able to lyse hematopoietic cells from HA-1+ patients with acute myeloid, T-, and B-cell lymphocytic leukemia (n = 15). No cytotoxic effect was observed on cells from HA-1- or HLA-A*02-negative donors (n = 10). The results support the use of HA-1 as a target for post-transplant T cell therapy.

## 1. Introduction

Allogeneic hematopoietic stem cell transplantation (allo-HSCT) is widely used for the treatment of acute myeloid leukemia (AML) and lymphoblastic leukemia (ALL) [1,2]. The therapeutic effect of allo-HSCT is largely mediated by the graft-versus-leukemia (GVL) response, in which residual recipient malignant cells are eliminated by the donor lymphocytes [3,4,5]. However, a significant proportion of the patients experience disease relapse [6,7,8,9,10]. Anti-relapse therapies include tyrosine kinase inhibitors, monoclonal antibodies, and antibody-based conjugates specific for CD19, CD20, CD22, and CD52 for lymphoblastoid leukemias; CD33, CD123, CD13, CLL-1, and CD38 [11,12] for myeloid malignancies [11,12]. However, pharmaceutical interventions remain challenging due to low efficacy and adverse side effects [13]. Patients with relapsed and refractory AML have a particularly poor prognosis [14]. Chemotherapy is still the most common treatment for relapsed and refractory AML [15]. A combination of venetoclax and hypomethylating agents was beneficial in the relapsed/refractory AML although the effect of such therapy is transient and it is best used as bridge therapy before HSCT [16,17]. Therapies targeting to IDH1/2 or FLT3 have significantly expanded the arsenal of treatment options, but they are still not curative [18,19]. Among antibody-based therapies, gemtuzumab ozogamicin had the highest efficacy, but its approval was withdrawn due to toxicity [20]. Therefore, allo-HSCT remains the most reliable option for patients with AML. Patients who relapse after transplantation are particularly in need of novel therapies [21].

Donor and recipient cells could be distinguished by their cell surface presentation of minor histocompatibility antigens (MiHAs), the peptides derived from proteins with polymorphic amino acids and presented by HLA molecules [22,23]. The donor immune response directed against the recipient MiHAs expressed predominantly or exclusively in the hematopoietic tissue could result in a beneficial GVL response without graft-versus-host disease (GVHD) [22,24].

The minor histocompatibility antigen HA-1 is a promising target for several reasons. First, it is presented by the HLA allele (HLA-A*02:01) common in the Caucasian population [25]. Second, its encoding gene ARGHAP45 (HMHA1) is exclusively expressed in hematopoietic tissue, including hematological malignancies [26,27,28]. HA-1 is derived from a histidine-encoding allelic variant (VLHDDLLEA) of the polymorphism rs1801284 [25]. Its arginine-encoding allelic counterpart VLRDDLLEA is non-immunogenic due to insufficient binding affinity with HLA-A*02:01 [29]. The genotype frequencies of rs1801284 for the A/A, A/G (immunogenic) and G/G (non-immunogenic) HA 1 variants are 16%, 36%, and 48%, respectively. Therefore, approximately half of the allo-HSCT recipients carry at least one HA-1 allele resulting in 25% of transplants being mismatched by this antigen [27]. It has been observed that recipients receiving HA-1-mismatched grafts in HLA-A*02-matched transplantations have lower relapse rates compared to the patients with HA 1 matched pairs [30,31]. Furthermore, the presence of HA-1-specific T cell clones after infusion of the donor lymphocytes was associated with a better outcome [32,33]. Therefore, the therapeutic strategy based on the adoptive transfer of HA 1 specific T cells could potentially be used as a targeted method to eradicate the residual disease while sparing the healthy hematopoietic system of the donor origin and non-hematopoietic tissues of the patient [22,24]. To generate HA-1-specific T cells, donor CD8^+^ T cells could be modified by lentiviral transduction to express a high-avidity transgenic HA-1-specific T cell receptor (TCR) [34,35].

In this study, we report the development of HA-1-specific T cell immunotherapy. Using the Jurkat J76 reporter T cell line [36], we have determined a set of functional HA-1-specific TCRs, estimated their affinity, and investigated their cross-reactivity against the panel of peripheral mononuclear blood cells (PBMC) with common HLA alleles. Three TCRs selected for their high affinity and the lack of cross-reactivity showed a response to PBMC with endogenously processed HA-1 peptide.

Endogenous TCR was knocked out by CRISPR/Cas in primary CD8^+^ T cells; a selected HA-1-specific TCR was then introduced by lentiviral transduction. Such HA-1-specific CD8^+^ T cells showed in vitro cytotoxicity against blood cells from HA-1–positive leukemia patients, but not against cells from HA-1 or HLA-A*02-negative patients. The described pipeline for the selection of the HA-1-specific TCRs and the production of the MiHA-specific CD8^+^ cells could be applied to a wide variety of different MiHAs presented by the other HLA alleles.

## 2. Materials and Methods

### 2.1. Patients and Samples

The study was approved by the Research Ethics Committee of the National Research Center for Hematology (Protocol № 126, 25 February 2022). All donors and patients gave informed consent before enrollment. Blood samples were collected during the patients’ hospitalization. Peripheral mononuclear blood cells (PBMC) were isolated from a whole blood sample by Ficoll gradient centrifugation (Paneco, Moscow, Russia). PBMC were stored frozen in FBS (Gibco, Paisley, UK) in the presence of 7% DMSO at −80 °C.

DNA from the blood samples was extracted using QIAamp DNA Blood Mini kit and QIAcube system (Qiagen, Hilden, Germany) according to the manufacturer’s protocol. HLA typing was performed either by next-generation sequencing (NGS) [37] or by flow cytometry (Appendix A) [38] (HLA-A*02 status only). The method used for the HLA testing for each sample is listed in Appendix A. Samples that were determined to be HLA-A*02-positive by flow cytometry but did not have HLA-A*02:01 allele, were discriminated by the J76 stimulation assay (Materials and methods, determining functionality and affinity of HA-1-specific TCR) and excluded from the analysis. The NGS libraries were prepared using AllType NGS amplification kits (One Lambda, Los Angeles, CA, USA) and sequenced using MiSeq Reagent Kit v2 (Illumina, San Diego, CA, USA). HLA genotyping was performed using the TypeStream Visual Software v2.0.0.68 (TSV) (One Lambda, Los Angeles, CA, USA) and the IPD-IMGT/HLA 3.40.0.1 database [39]. HA-1 genotyping was performed as previously described [40].

### 2.2. Generation of HA-1-Specific T Cell Clones

Naive CD8^+^ T cells from HA-1 donors (rs1801284 G/G) were isolated from PBMC using a naive T cell immunomagnetic isolation kit (130-045-201, Miltenyi Biotec, Bergisch Gladbach, Germany) according to the manufacturer’s protocol. Cells were cultured in RPMI medium (Gibco, Paisley, UK) supplemented with 10% FBS, 50 U/mL IL-2 (Biotech, Moscow, Russia), 0.5 U/mL IL-7 and 0.8 U/mL IL-15 (130-095-362 and 130-095-765, Miltenyi Biotec, Bergisch Gladbach, Germany), and 100 U/mL penicillin/streptomycin (Gibco, Paisley, UK). For the isolation of dendritic cells (DCs), monocytes from the same donor were purified using anti-CD14 immunomagnetic beads (130-050-201, Miltenyi Biotec, Bergisch Gladbach, Germany) and cultured in RPMI medium supplemented with 10%FBS, 100 U/mL penicillin/streptomycin, 800 U/mL GM-CSF (130-093-864, Miltenyi Biotec, Bergisch Gladbach, Germany), 100 U/mL IL-4 (130-095-765, Miltenyi Biotec, Bergisch Gladbach, Germany), 10 ng/mL lipopolysaccharide (L2630-10MG, Sigma Aldrich, Taufkirchen, Germany), and 100 U/mL IFN-γ (130-093-864, Miltenyi Biotec, Bergisch Gladbach, Germany). After 3 days, DCs were detached with a cell scraper and irradiated for 50 min with a total dose of 50 Gy using Biobeam GM 8000 (Gamma-Service Medical, Leipzig, Germany). Irradiated DCs were pulsed with synthetic HA-1 peptide (LifeTein, Somerset, USA) at a final concentration of 5000 ng/mL (4.88 nmol/mL).

Naive CD8^+^ T cells were co-cultured with the irradiated DCs of the same donor (or, in the case of HLA-A*02:01- donors, with allogeneic DCs from HLA-A*02:01^+^ donors) for 10 days as described previously [41].

Briefly, cells were seeded in 48-well suspension plates at a density of 2 × 10^5^ to 1 × 10^6^ per well in RPMI media supplemented with 10% FBS, 100 U/mL penicillin/streptomycin, 30 ng/mL IL-21 (130-095-784, Miltenyi Biotec, Bergisch Gladbach, Germany). The ratio of plated CD8^+^ naive T cells to DC cells was 2:1 or 4:1. Both cytokines IL-7 and IL-15 (Miltenyi Biotec, Bergisch Gladbach, Germany) were added at concentrations of 100 U/mL each on days 3, 5, and 7. After co-culture, the T cell cultures were screened for antigen specificity by restimulation followed by CD137 or direct tetramer staining as described in the Appendix A (CD8^+^ T cell activity assays and flow cytometry analysis).

### 2.3. HA-1-Specific TCR Repertoire Sequencing and Bioinformatic Analysis

Tetramer^+^ or CD137^+^ cells of T cell expansions were selected using anti-PE immunomagnetic beads (Miltenyi Biotec, Bergisch Gladbach, Germany), and RNA was purified using RNEasy mini columns (Qiagen, Hilden, Germany) followed by cDNA library synthesis as described previously [42]. Briefly, a universal primer specific for the α or β TCR constant region was used to prime a reverse transcription reaction using high precision Moloney murine leukemia virus reverse transcriptase (SMARTScribe, Takara, Kusatsu, Japan), which proceeds according to the method of rapid amplification of complementary DNA (cDNA) 5′ end (5′RACE) [43]. After reverse transcription, a unique molecular identifier (UMI) and a sample barcode were inserted at the 3′ end of the first cDNA strand via a template switch. Next, the cDNA chains were amplified using two-step nested PCR. In the second step of the nested PCR, Illumina sequencing adaptors were introduced. α and β TCR libraries were generated separately from a single cDNA library. The PCR libraries were then sequenced using the Illumina MiSeq Kit v2 and the MiSeq system (Illumina, San Diego, CA, USA). NGS data were analyzed using the MiXCR and VDJtools frameworks [44,45]. The number of reads of each CDR3 sequence in the antigen-specific fraction and flowthrough was compared for statistically significant (exact Fischer test, *p* < 0.05) and strong (10-fold or greater) enrichment. α- and β-chains enriched in CD137^+^ or tetramer^+^ fractions from the same expansion well were used for cloning into lentiviral modules as described below.

### 2.4. Determining Functionality and Affinity of HA-1-Specific TCR

For transgenic TCR reactivity assays, the J76 reporter cell line was used [36]. This cell line lacks endogenous TCR and is engineered to express fluorescent proteins under promoters activated by TCR signaling. In our readouts, we measured the GFP expression driven by the NFAT promoter. To increase the sensitivity of the system, we generated a J76 cell expressing transgenic CD8, CD2, and CD28 by lentiviral transduction (see Appendix A, lentiviral transduction of cell lines). K562 cells (ATCC CCL-243™) with transgenic HLA-A*02:01 were used for activation assays.

J76 cells harboring a transgenic TCR were prepared as described (Materials and methods, Transgenic TCR assembly, and Appendix A, Lentiviral transduction of cell lines). For the TCR stimulation assay, 5 × 10^5^ of the obtained TCR transgenic J76 cells were incubated overnight with 1 × 10^6^ K562-HLA-A*02 cells that were pulsed with HA-1 peptide to a final concentration of 4,88 nmol/mL. Activation of J76 cells was determined by GFP expression using flow cytometry (Figure 1C). GFP-positive J76 cells were considered to have functional HA-1-specific TCR. Cell lines with functional TCRs were enriched by immunomagnetic separation for CD3 (130-050-101, Miltenyi Biotec, Bergisch Gladbach, Germany) and then sorted using anti-TCRα antibody (Sony, Tokyo, Japan, 2133590) staining on the BD FACS Aria III cell sorter. The sorted J76 cell lines were used to measure TCR affinity. J76 cells, previously labeled with lipophilic DID stain (Invitrogen, Waltham, USA), were stimulated with 10 fivefold serial dilutions of HA-1 peptide, starting with the highest concentration of 2.5 × 10^4^ ng/mL (24.4 nmol/mL). We co-cultured 1.25 × 10^5^ of J76 cells with K562-A*02:01 cells in a 1:2 ratio with an indicated amount of peptide in a 96-well plate in duplicate. After overnight stimulation, the percentage of GFP-expressing DID-positive J76 cells was analyzed using a MACS Quant flow cytometer (Miltenyi Biotec, Bergisch Gladbach, Germany). EC50 values for the plotted TCR titration curves were estimated using GraphPad Prizm 9 software (GraphPad, San Diego, CA, USA).

Stimulation of sorted J76 cells by PBMC from healthy donors and leukemia patients was performed as follows: PBMC were seeded in a 96-well plate at a density of 2.5 × 10^5^ cells/well and 1.25 × 10^5^ DID-labeled J76 cells were added. Stimulation was performed in triplicate. As a positive control for each triplicate, exogenous HA-1 peptide at a concentration of 4.88 nmol/mL was added to the fourth well. After overnight cocultivation at 37 °C and 5% CO_2_, GFP expression was measured by flow cytometry using MACS Quant. Samples that were determined to be HLA-A*02 positive by flow cytometry but failed to elicit a J76 response in a positive control assay were excluded from the analysis as HLA-A*02:01 negative.

### 2.5. CD8^+^ T Cell Activity Assays

The cytotoxic assay to evaluate the functional activity of CD8^+^ T cells modified with transgenic TCRs was performed by monitoring caspase 3/7 levels by flow cytometry analysis and IFN-γ by ELISA.

For the caspase-killing assay, 2.5 × 10^5^ PBMC were seeded in a 48-well plate and 1.25 × 10^6^ DID-labeled effector CD8^+^ T cells were added. The experiment was performed in triplicate. As a positive control, 4.88 nmol/mL HA-1 peptide (LifeTein, Somerset, NJ, USA) was added and as a negative control, mock (PBS) transduced effectors were added. After overnight cocultivation at 37 °C and 5% CO_2_, cells were transferred to a round-bottomed 96-well plate, pelleted by centrifugation at 300× *g* for 5 min, and stained using the CellEvent Caspase-3/7 Green Flow Cytometry Assay Kit (eBioscience, San Diego, CA, USA) according to the manufacturer’s protocol. Plates were analyzed on a MACS Quant flow cytometer. The percentage of caspase-3/7 and 7AAD-positive events in the DID-negative fraction was measured to determine the cytotoxic effect.

The functional activity of TCR-modified CD8^+^ T cells was followed by the amount of IFN-γ produced in the media, measured by the IFN-γ ELISA assay (Hema, Moscow, Russia) according to the manufacturer’s protocol. Cells in the functional assay were stimulated with serial dilutions of HA-1 peptide as described above.

Expansions of CD8^+^ T clones were screened with irradiated B lymphoblastoid cells (B-LCL) from the same donor. The B-LCL cells (Appendix A, LCL generation) were irradiated (50 min, 5 Gy) and tested for antigen specificity loaded with 4.88 nmol/mL HA-1 peptide. CD137 expression was measured by flow cytometry after 16 h of stimulation (Appendix A, flow cytometry analysis).

### 2.6. Transgenic TCR Assembly

For rapid cloning of the desired TCR, we developed a modular Golden Gate assembly system (Figure 1B). Human constant TCR chains were synthesized by RT-PCR from mRNA isolated from the PBMC of a healthy donor. Murine constant TCR chains [46] were codon-optimized for human expression using the iCodon algorithm and synthesized by Evrogen LLC, Russia. To stabilize the recombinant TCR receptors, the cysteine substitutions were additionally introduced into all constant chains [47]. The TCR α and β chains were spanned by the P2A peptide to ensure chain separation during translation [48]. The variable α and β chains selected for TCR cloning were amplified using a pair of primers for specific V and J segments flanked by the *Bpi*I sites and the corresponding NGS libraries as templates. The PCR products were analyzed and purified by 1% agarose gel electrophoresis using the Gene Jet Gel Extraction Kit (Thermo Fisher Scientific, Waltham, MA, USA) and either used directly for Golden Gate assembly of the final lentiviral constructs or subcloned into the pJet1.2 plasmid (Thermo Scientific, Waltham, MA, USA). Unwanted *Bpi*I sites were removed from the variable chains by PCR mutagenesis.

The final plasmid was constructed from modules containing two variable and two constant chain regions and the lentiviral backbone vector by the Golden Gate assembly reaction [49] using *Bpi*I restriction endonuclease (Thermo Fisher Scientific, Waltham, MA, USA) and T4 ligase (Thermo Fisher Scientific, Waltham, USA). The Golden Gate assembly was performed by 9 to 30 cycles of 15 min restriction at 37 °C followed by 10 min ligation at 16 °C. The reaction was terminated by 5 min at 55 °C. The reaction mix was then transformed into NEB stable competent cells (NEB, Ipswich, USA), and transformants were screened by PCR. Plasmid constructs were isolated using the Gene Jet Miniprep kit (Thermo Fisher Scientific, Waltham, MA, USA) and sequenced using the SeqStudio Genetic Analyzer (Applied Biosystems, Waltham, MA, USA).

### 2.7. Lentiviral Transduction, Purification, and Expansion of Primary T cells

The purified primary CD8^+^ T cells were activated using T cell anti-CD3/2/28 activation/expansion kit (Miltenyi Biotec, Bergisch Gladbach, Germany) according to the manufacturer’s protocol, and further cultivated for 72h in the RPMI media supplemented with 10% FBS, 500 U of IL-2, 100 U of IL-15, and 100 U/mL penicillin/streptomycin. The non-treated 24-well culture plates (Sarstedt, Nümbrecht, Germany) were coated with 30 µg/mL retronectin (Takara, Kusatsu, Japan) and blocked with 2% BSA solution (Sigma Aldrich, Taufkirchen, Germany) at 37 °C and 5% CO_2_ overnight to be used for lentiviral transduction. Activated CD8^+^ T cells (1 × 10^6^/well) were seeded into retronectin-coated plates, and thawed lentiviral supernatant (Appendix A, *lentivirus manufacturing*) was added to achieve MOI~6. Plates were then centrifuged at 1000× *g* during 45 min at 32 °C and cultured at 37 °C and 5% CO_2_ for 72h. Cells expressing transgenic murine TCR were purified by staining with anti-mouse TCR β antibody, followed by the anti-APC immunomagnetic separation on beads (130-090-855, Miltenyi Biotec, Bergisch Gladbach, Germany), and further expanded for a total of 4–5 weeks with fresh medium changed every week.

### 2.8. CRISPR/Cas Knockout of Endogenous TCR

A total of 2.5 × 10^6^ activated T cells were electroporated with ribonucleoprotein (RNP) complexes (see Appendix A, *Ribonucleoprotein complexes*) using the Neon transfection system (Thermo Fisher Scientific, Waltham, MA, USA). We added 20 µL of RNP complexes to T cells in 90 µL of electroporation buffer T supplemented with 11 µM electroporation enhancer (IDT, San Diego, MA, USA). Electroporation pulse parameters were 1600 V, 10 ms, 3 pulses. If further lentiviral transduction was required, T cells were transferred to retronectin-coated plates with virus-containing medium immediately after electroporation and were then proceeded to spinfection as described previously. Alternatively, T cells were immediately transferred into 1.5 mL of pre-warmed medium in a 24-well plate and incubated for 72 h at 37 °C and 5% CO_2_ for further flow cytometric analysis to determine TCR knockout efficacy. For the gRNA efficacy screening, Jurkat E6-1 cells (ATCC TIB-152™) were electroporated as described above.

## 3. Results

### 3.1. Bioinformatic Analysis Revealed the Low Degree of CDR3 Similarity in HA-1-Specific TCR Repertoire and Predominant Usage of TRBV7-9 Gene

We expanded naive CD8^+^ T cells of five healthy HLA-A02:01+ HA-1 donors. We complemented them with expansions of T cells from three HLA-A02:01 donors (Figure 1A and Appendix A), as allogeneic expansions have been previously reported to be a source of TCRs [50,51]. The antigen-specific T cells were detected in 16–50% of the individual expansion wells (Appendix A), indicating that the naive HA-1−specific T cells were relatively rare. TCRs were sequenced in the flow-through and antigen-specific fractions (Materials and methods, Appendix A). Sequencing revealed that 50 α and 68 β chains were strongly (10-fold or higher) and significantly (exact Fischer test, *p* < 0.05) enriched in the antigen-specific fraction, indicating they belonged to the HA-1–specific TCRs (Appendix A).

The CDR3 regions discovered here, plus previously published [35,52] HA-1−specific α and β TCR chains, were clustered according to the Levenshtein distance (Figure 2A). The analysis showed a low level of homology; the majority of CDR3 sequences were unique and did not belong to any homology cluster.

*TRBV7–9* was the most abundant V-gene among TCR β chains (Figure 2B), suggesting the importance of its C-terminal amino acids for HA-1 peptide recognition. No significant bias in V-gene usage was found among α chains. Analysis of V and J gene combinations did not reveal any V–J pairs that were significantly more frequent than others. Compared to TCR repertoires specific for the well-studied epitopes of cytomegalovirus (KLG and NLV) and Epstein–Barr virus (AFV) (data from VDJdb), TRBV genes of the HA-1−specific TCR repertoire had significantly lower diversity (Shannon diversity index 1.6) (Appendix A), and a similar low diversity was observed in the GIL-specific repertoire (Influenza A). For TRAV genes, the diversity index did not differ significantly between the repertoires (Shannon diversity 3.3).

The CDR3 length varied from 8 to 19 and 11 to 17 amino acids for α and β chains, respectively (Appendix A). The length distribution was normal with a mode of 13 amino acids for β chains (n = 35) and 14 amino acids for α chains (n = 14).

### 3.2. J76 Cell Reporter Assay Revealed Functional Chain Combinations and Estimated the Avidity of HA-1-Specific TCRs

We cloned 25 β and 29 α bioinformatically discovered TCR chains and assembled 48 recombinant TCRs combining chains obtained from the same expansion well (supplementary Appendix A). Using the J76 reporter cell line [36], we identified 13 functional HA-1-specific TCRs. The EC_50_ values calculated from the peptide–titration curves (Appendix A) are shown in Figure 3A. The measured affinities of the TCRs mostly varied in the typical for TCR range of 10–100 µM [53]. Staining with recombinant pMHC dextramer showed some discrepancy with the results of the functional titration assay (Figure 3B). For example, ER6 and ER8, both determined to have high affinity in the functional titration assay, showed low levels of dextramer binding. The medium affinity receptor ER29 and the low affinity receptors ER4, PKS3, ER17, and ER23 showed no dextramer binding. However, it is known that functional avidity is a better representation of TCR activity and does not correlate uniformly with the affinity of the TCR to the peptide–MHC complex [54]. Surprisingly, some TCRs derived from the tetramer-enriched population (ER17, ER29) failed to bind dextramer when expressed in J76 cells. The most plausible explanation could be the different expression levels of transgenic TCR and CD8 expression in J76 cells compared to primary CD8^+^ T cells.

TCR PKS11 obtained from the allogeneic expansion showed the highest level of dextramer binding. However, functional titration showed its activation even at the lowest peptide concentration (Appendix A), suggesting peptide-independent HLA recognition. Stimulation with healthy donor PBMC confirmed that PKS11 was alloreactive to HLA-A*02:01 (Appendix A). For further analysis, we selected three TCRs that showed both high affinity in titration assays, high dextramer binding, and were not alloreactive to HLA-A*02: ER6, ER12, and ER28.

### 3.3. Selected Transgenic TCRs Specifically Recognized Endogenously Processed HA-1 Peptide

The transgenic J76 cells modified with the ER6, ER12, and ER28 TCR receptors were stimulated by PBMC from healthy donors or leukemia patients with HLA-A*02:01^+^ HA-1^+/+^, HA-1^+/−^, HA-1^−/−^ and HLA-A*02:01^−^ (Figure 3C,D). PBMCs from the majority of HA-1^+/+^ and HA-1^+/−^ healthy donors were able to elicit a reporter cell line response, whereas no activation was observed upon stimulation by the cells from HA-1^−/−^ or HLA-A*02:01^−^ donors (Figure 3C). HA-1^+/+^ healthy donor cells activated a higher percentage of reporter T cells compared to HA-1^+/−^ cells, which probably could be explained by higher abundance of HA-1-HLA complexes on the cell surface. No such difference in reporter activation was observed for PBMC from leukemia patients (Figure 3D). Furthermore, PBMC from leukemia patients were found to have lower HLA-A*02:01 expression levels than PBMC from healthy donors (Appendix A); downregulation of HLA expression is one of the mechanisms of tumor immune evasion [55]. At the same time, we did not observe any obvious correlation between the diagnosis or the percentage of blast cells in the sample and the level of reporter activation (Appendix A).

Of the three HA-1-specific receptors tested, ER28 demonstrated the highest level of activation upon antigen stimulation. To confirm the absence of cross-reactivity, we incubated J76 reporter cells with PBMC samples from 21 healthy donors with the most frequent HLA alleles (Appendix A). Neither cell line was activated by PBMC from 21 donors that were negative for HA-1 or HLA-A*02:01 (Appendix A).

### 3.4. CD8^+^ T cells with Murinized Transgenic HA-1–Specific TCRs and CRISPR/Cas Knockout of Endogenous TCR Showed Specific Lysis of PBMC from HA-1^+^ Patients with Various Hematological Malignancies

Murinization of transgenic TCR constant chains and cysteine modification are widely used in T cell therapy; they increase the expression level of transgenic TCR and allow direct measurement of transduction efficiency by flow cytometry analysis [46,56,57]. Primary CD8^+^ T cells from an HLA-A*02:01^−^ donor were transduced with three murinized HA-1-specific TCRs; transgenic TCR expression in the cells with intact endogenous TCR as assessed by flow cytometry was 8–9% for all three cultures (Appendix A). Transduced cultures secreted IFN-γ in a dose-dependent manner upon exogenous HA-1 peptide stimulation, with cultures carrying each of the three TCRs secreting the same amount of IFN-γ at the maximum peptide concentration (Figure 4A). Considering the results of dextramer staining, peptide titration experiments, and stimulation with donor and patient-derived PBMC, the TCR ER28 was selected as the most promising TCR candidate for further analysis.

First, we compared the knockout efficiency of previously published gRNAs and a set of gRNAs provided by the Synthego CRISPR design tool (Appendix A) [58,59]. Published gRNAs demonstrated the most efficient TCR knockout in the Jurkat E6-1 cell line (Appendix A) and primary CD8^+^ T cells from healthy donors, reaching 80–90% knockout efficiency (Appendix A).

Second, TCR knockout increased the MFI of HA-1-HLA-A*02:01 dextramer staining of primary CD8^+^ T cells transduced with HA-1-specific TCR ER28 by 40–90% (Figure 4B). The increase in dextramer binding intensity after TCR knockout suggests that transgenic TCR expression was enhanced due to reduced competition with endogenous TCR chains.

Finally, we demonstrated the cytotoxic effect of HA-1 TCR-transgenic CD8^+^ T cells on PBMC from leukemia patients. CD8^+^ T cells from two HLA-A*02:01^−^ donors were subjected to endogenous αβ TCR knockout, transduced with ER28 TCR containing murine constant chains, magnetically sorted for murine TCR expression, and expanded more than tenfold (Appendix A). HA-1 dextramer staining after two weeks in culture showed that > 90% of cells in enriched cultures were HA-1-specific (Appendix A). HA-1-specific CD8^+^ T-lymphocytes exhibited marked cytotoxicity against PBMC from leukemia patients with HA-1^+/+^ and HA-1^+/−^ genotypes, while no cytotoxic effect was observed against cells without HA-1 or HLA-A*02:01 expression (Figure 4C,D). The addition of exogenous HA-1 peptide increased the killing, confirming that the cytotoxic response was directed against the HA-1 peptide (Figure 4E). While the majority of PBMC samples belonged to patients with AML (Appendix A), the cytotoxic effect was also demonstrated in other diagnoses and was independent of blast percentage (Appendix A), supporting the evidence that HA-1 is a universal hematological target suitable for immunotherapy of a broad range of hematological malignancies.

## 4. Discussion

Among all types of T cell immunotherapy, chimeric antigen receptor T cells (CAR-T) currently dominate, with more than 600 active clinical trials underway [60] and more than 80 potential targets identified [61]. Compared to CAR-T, TCR-T therapy is still in its infancy, with approximately 100 clinical trials targeting a total of 19 antigens [62,63]. This discrepancy could be explained by the complexity of TCRs and their antigen identification, which requires a laborious ex vivo culture of T cells, their expansion, and subsequent analysis of the TCR repertoire [64]. However, the therapeutic application of CAR-T is limited for certain diagnoses due to the lack of suitable cell surface targets. The therapy-induced myelotoxicity limits the use of CAR-T to a bridging therapy prior to allo-HSCT [65]; therefore, other therapeutic approaches are needed [66].

Another area where TCR-T has some advantages over CAR-T is in the treatment of solid tumors [67]. The factors that complicate the use of T cell therapy in solid tumors are poor tumor infiltration, immunosuppressive microenvironment, and tumor antigen heterogeneity [68]. Many surface antigens are not essential for cancer progression and their expression could be easily ablated by cancer cells, or the targetable antigens are expressed at low levels; in addition, most antigens available for CAR-T are TAAs, which significantly increases the chance of on-target off-tumor toxicity [69]. TCR-T therapy has the advantage of targeting a diverse set of solid tumor antigens: in contrast to the CAR-T approach, which exclusively targets surface molecules, TCR-T could potentially target any protein if its peptides are presented by the appropriate HLA [70]. Thus, the patient-specific TCR repertoire of tumor-infiltrating lymphocytes could be used to advantage in TCR-T therapy [71]. As shown in a recent study, claudin-specific CAR-T cells could be efficiently stimulated by an mRNA vaccine that induced dendritic cells to express CLDN6 on their surface [72]. The same strategy could be applied more efficiently with TCR-T therapy, since TCR antigens are much smaller and can be delivered by peptide, mRNA, or oncolytic virus vaccine [73]

The efficiency of antigen presentation by HLA is the key factor to consider in the design of an efficient TCR-T therapy. In addition, the immunogenicity of the peptide and the differences in the expression pattern of the antigen source in healthy and tumor cells are two other factors influencing the effect of TCR-T therapy. These limitations drastically limit the choice of antigens: the majority of TCR-T therapies use the NY-ESO-1 and WT-1 targets against solid and hematological malignancies, respectively [63]. The high immunogenicity of NY-ESO-1 is rather an exception [74]. In contrast, tumor neoantigens are more immunogenic; only a few are shared by a large number of patients [75]. Therefore, the search for novel and more immunogenic peptides derived from TAAs, as well as the enhancement of TCR affinity [76] are required for the improvement of new generations of the TCR-T therapy in the future.

The use of MiHAs as immunotherapy targets may be more advantageous in the context of allo-HSCT therapy than targeting the TAA and tumor neoantigens. MiHAs are more immunogenic than TAA because they are completely foreign to the MiHA-negative immune system of the donor [77]. In addition, MiHAs are much safer targets because they have less on-target off-tumor toxicity [78]. MiHA-targeted therapy is not patient-specific compared to the majority of tumor neoantigens. Good targets for MiHA-targeted therapy can be identified without extensive tumor genome and transcriptome sequencing [79]. In addition, when MiHAs arise from germline polymorphisms in genes whose function is important to the malignant cell, tumors are less likely to escape the MiHA-directed immune response due to the downregulation of the source gene. Potential tumor escape is most likely due to loss of HLA [80]. This is particularly important in the treatment of AML, as this type of leukemia is thought to have many subclones at baseline, and targeting a germline polymorphism such as MiHA seems to be an appropriate strategy [80]. Therefore, the transfer of T cells modified with the transgenic TCRs seems to be a promising method of immunotherapy to complement allo-HSCT for efficient relapse prevention, applicable to the treatment of a wide spectrum of hematological malignancies [81,82].

Currently, only a few clinical trials are focused on the use of MiHA-specific transgenic T cells for the treatment of relapsed and refractory hematologic diseases, especially AML [34]. Taking into account the frequency of HLA and MiHA variants, it could be estimated that the development of 50 MiHA-based therapies would be sufficient to effectively treat 35% of patients undergoing allo-HSCT [83].

The pipeline of the antigen-specific expansion and TCR discovery outlined in our work (Figure 1) could be efficiently applied to generate the TCR repertoires specific to the other therapeutically promising MiHAs such as HA-2 [84] and ACC-1Y [85], as well as any other T cell antigen. This study is the first systemic analysis of an MiHA-specific TCR repertoire, combining our newly generated data with those published previously [34,35]. Our results differ from the previously published data on well-studied virus-specific TCR repertoires, such as SARS-CoV2 and CMV. The HA-1-specific TCR repertoire demonstrated a low degree of overall sequence homology and low diversity of the β V-genes with the TRBV7–9 gene, being the most commonly used. Analysis of the antigen-specific repertoire may be useful for design of TCR-T therapy; some of the identified V-genes were reported to be “weak” due to less efficient folding, which affects the transgenic TCR exposure on the cell surface [86].

The efficient expression and exposure of the transgenic TCR could be improved not only by the choice of the V-gene, but also by the use of the murine TCR constant chains [46,59]. Inevitably, this approach raises concerns about the safety and immunogenicity of the T cell product, as foreign parts of the TCR could trigger an immune response [87]. Indeed, clinical trials showed that the murine TCR-specific antibodies were detected in approximately 23% of patients after infusion of the autologous T cells modified with the murinized transgenic TCR [57]. However, the generated antibodies were neutralizing in only half of the reported cases, which could have affected the efficacy of the therapy. More importantly, the antibodies generated after transfer were specific for a variable part of the TCRs exposed from the cell surface and more accessible for immune recognition. We used only constant TCR chains of murine origin in our constructs to minimize the immunogenicity of the transgenic TCRs. However, there is still the possibility of the recipient’s CD8^+^ T cells developing an immune response to the peptides derived from the foreign sequences of the genetic construct [88]. While little is known about CD8^+^ responses directed against murinized TCR, there is evidence that such responses impair the efficacy of CAR-T therapy [89,90]. Therefore, elimination of T cell-immunogenic epitopes of the murine TCR may be required for improved therapeutic efficacy.

The generated transgenic TCR-T cell products pose a potential safety risk because transgenic and endogenous TCR chains could form heterodimers of unknown reactivity [91]. To circumvent this problem, the CRISPR/Cas knockout of endogenous TCR could be used [92,93] in addition to the introduction of murine constant TCR chains. Genetic modification of cells by electroporation of RNP complexes proved to be safer compared to lentiviral methods of CRISPR/Cas delivery. The Cas9 protein did not elicit an immune response due to its short-lived presence in the organism. The off-target activity of Cas9 was shown to be insignificant to undermine the genetic stability of the T cell therapeutic product [94]. Moreover, the increasing commercial availability of Cas9 protein and the upcoming availability of clinical-grade electroporation systems are likely to make CRISPR/Cas disruption of the endogenous TCR a standard procedure for the TCR T cell product manufacturing.

The persistence of modified T cells is essential for therapeutic efficacy [70]. CAR-T cells with 4–1BB costimulatory domain showed better persistence than CAR-T with CD28 domain, which is attributed to a more moderate level of receptor activation and less exhaustion [95]. Transgenic TCR induces physiological levels of cell activation compared to CAR, and a direct comparison of CAR-T and TCR-T revealed that although CAR-T are more potent effectors in the short term, TCR-T cells perform better under high antigenic pressure, showing less exhaustion and expanding more efficiently [96]. The reported in vivo persistence of TCR-T cells that underwent ex vivo expansion and adoptive transfer could vary from 1 week for expanded tumor-infiltrating lymphocytes [97] to more than 430 days in some patients [98]. Transgenic TCR-T have been reported to persist for at least 1–2 months [99,100] and were detectable for more than 6 months in some cases [101]. The use of high doses of IL-2 during ex vivo culture and the resulting effector memory phenotype of the infused cells are considered to be the main factors that negatively influence the persistence of the transferred cells and the efficacy of the therapy, as concluded from the HA-1-specific adoptive transfer clinical trials [102]. The strategy to increase the persistence of CAR-T by modification of naïve and stem-cell memory populations [103] may also be applicable to TCR-T. In some cases, CAR-T persistence reached 10 years after infusion, and it is reasonable to expect no less persistence from TCR-T [104].

Advanced methods of gene engineering and novel bioinformatic tools for peptide immunogenicity prediction have great potential to make TCR T therapy a highly efficient and specific method of choice when other approaches fail, particularly for the therapy of relapsed and refractory AML. MiHAs are particularly promising targets for this purpose because, unlike other classes of antigens, they allow efficient discrimination between donor and recipient cells.

## 5. Conclusions

We reported the experimental pipeline for the development of MiHA HA-1 specific TCR-T therapy. We described the repertoire of HA-1-specific TCRs, identified a number of functional/TCR chain combinations, and then selected several TCR receptors with sufficient affinity and no alloreactivity. The ability of the cloned TCRs to effectively recognize endogenously processed HA-1 antigens on the surface of cells derived from healthy and leukemia patient PBMCs was demonstrated. We proposed a method for rapid TCR cloning and demonstrated its use for subsequent modification of CD8^+^ T cells with HA-1-specific transgenic TCRs after CRISPR/Cas knockout of an endogenous TCR. The resulting T cells showed clear cytotoxic activity against PBMC of patients with various hematological malignancies, including AML, B-, and T-cell ALL. The described pipeline could be applied to the development of novel therapies targeting other minor histocompatibility antigens (MiHAs), which represent a promising class of antigens for the treatment of hematological malignancies.

## Figures and Tables

**Figure 1 cancers-15-01592-f001:**
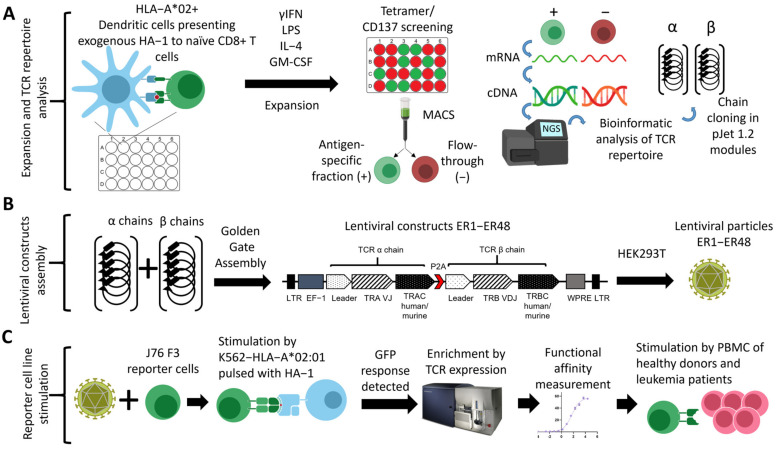
Schematic workflow of the study. (**A**) Naive CD8^+^ T cells were expanded using autologous or allogeneic dendritic cells from the HLA-A*02^+^ donors. Individual wells were screened for the presence of antigen-specific cells by tetramer or CD137 staining. The HA-1−specific TCR chains were identified after sequencing and bioinformatic analysis of the TCR repertoire of tetramer/CD137^+^ fraction and flowthrough. Selected HA-1−specific TCRs α and β chains were cloned into pJet 1.2 plasmids and used as the modules for the seamless cloning of lentiviral constructs. (**B**) Using a lentiviral backbone specifically designed for seamless cloning, the lentiviral constructs containing 48 TCR a/b chain combinations were assembled and used to produce lentiviral particles containing. (**C**) The reporter J76 cell lines expressing transgenic TCRs were generated; the functionality and affinity of each TCR were assessed by peptide stimulation. The response of cell lines with TCR receptors ER12, ER6, and ER28 was tested on PBMCs from healthy donors and leukemia patients.

**Figure 2 cancers-15-01592-f002:**
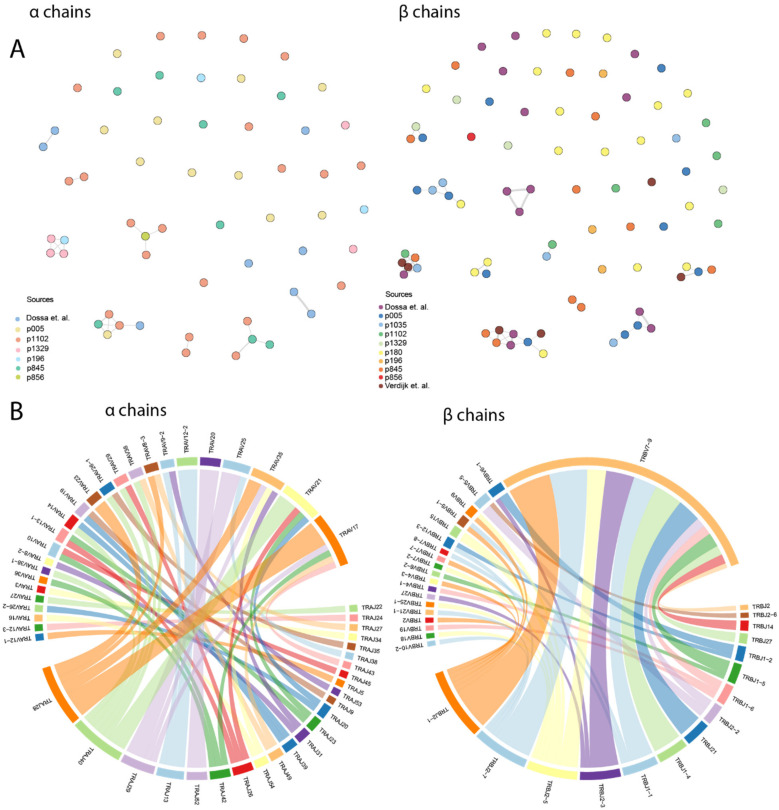
Bioinformatic analysis revealed the low degree of CDR3 similarity and the bias in usage of *TRBV7-9* gene in the HA-1-specific TCR repertoire (**A**) Levenshtein distances between the CDR3 amino acid sequences of HA-1-specific TCR α and β chains. Each chain is represented by colored circle, connecting lines represent 1–3 amino acid substitutions. (**B**) Frequency analysis of V and J gene usage showed that *TRBV7-9* is overrepresented in the repertoire of β chains, whereas V genes in the α chain repertoire are more equally distributed. Entanglement analysis did not reveal any bias in the V and J genes pairing [35,52].

**Figure 3 cancers-15-01592-f003:**
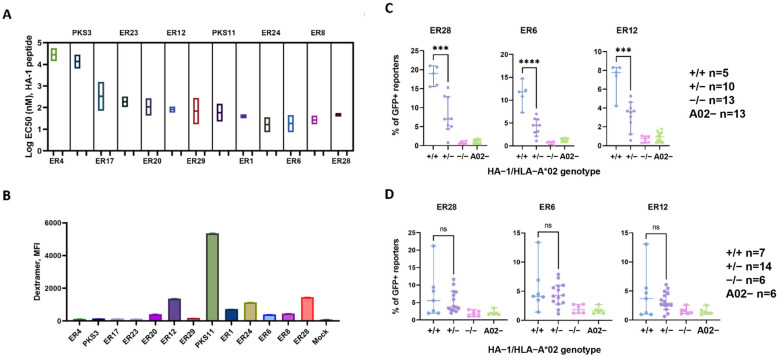
J76 cells modified with HA-1 − specific TCRs showed a response to the exogenously and endogenously processed HA-1 peptide and were stained with HA-1 HLA-A*02:01 dextramer. (**A**) J76 cell lines carrying functional HA-1 specific TCRs were stimulated by HLA-A*02:01^+^ K562 cells loaded with different concentrations of HA-1 peptide as described in Materials and methods. Mean values of log_10_ (EC_50_) for each TCR were obtained from two independent peptide titration experiments. (**B**) The J76 reporter cell lines staining with HA-1 HLA-A*02:01 dextramer. MFIs from three independent stainings are shown; error bars represent mean absolute error. (**C**,**D**) The J76 reporter cells expressing transgenic TCRs were stimulated by PBMC from healthy donors (**C**) and leukemia patients (**D**). Each dot represents the mean percentage of the GFP^+^ reporter cells from three independent stimulation experiments using cells of a single donor with an indicated genotype. Statistical significance: unpaired *t*-test *** *p* < 0.005, **** *p* < 0.0001.

**Figure 4 cancers-15-01592-f004:**
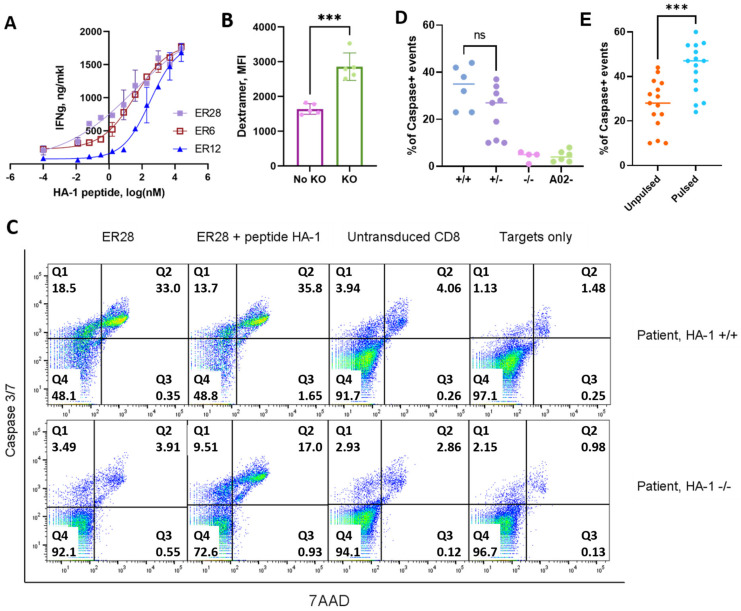
CD8^+^ T cells with murinized transgenic HA-1–specific TCRs and CRISPR/Cas knockout of endogenous TCR showed dose-dependent INF-γ response and specific lysis of PBMC from HA-1+ patients with various hematological malignancies. (**A**) CD8^+^ T cells modified with transgenic HA-1 specific TCRs stimulated by HLA-A*02^+^ K562 and exogenous HA-1 peptide. (**B**) The HA-1 HLA-A*02:01 dextramer staining of the CD8^+^ T lymphocytes transduced with HA-1−specific TCR receptor ER28, without and with CRISPR/Cas9 knockout of endogenous TCR; the data for five donors is shown; bars represent mean absolute error. (**C**) PBMCs from leukemia patients were cultivated with the ER28 transgenic CD8^+^ T cells after endogenous TCR knockout. The ER28 transgenic CD8^+^ T cells were cultivated with PBMC pulsed with 4.88 nmol/mL of HA-1 peptide as a positive control. The mock-transduced (PBS) CD8^+^ T cells were used as a negative control. The patient PBMC cultured without effectors were used as control of background cell death. Plots of caspase 3/7 and 7AAD staining are shown for two representative patients. (**D**) The ER28 transgenic CD8^+^ T cells demonstrate specific lysis of the PBMC from HA-1^+^ patients. No cytotoxicity to the cells of HA-1^−^ and HLA-A*02^−^ donors were observed. Each dot represents the mean percentage of the Caspase/7AAD positive events observed by flow cytometry analysis of two independent cultures of CD8^+^ T cells from two different donors. (**E**) Transgenic CD8^+^ T cells demonstrate a higher level of cell killing when exogenous HA-1 peptide was added compared to samples where only endogenously processed peptide was present. Statistical significance: unpaired *t*-test *** *p* < 0.001.

## Data Availability

The data presented in this study are available in this article.

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
