# Peer review of "Transgenic HA-1-Specific CD8+ T-Lymphocytes Selectively Target Leukemic Cells"

_cancers, 2023, doi:10.3390/cancers15051592_

Round 1
Reviewer 1 Report
The authors generated transgenic HA-1-specific CD8+ T-lymphocytes and showed that these CD8+ cells are able to kill leukemic cells in vitro. This study is a good model of graft versus leukemia effect and may lead to clinical application. This study has been reasonably performed and manuscript preparation is generally excellent.
Major comments:
1. Please show the expansion efficiency of HA-1-specific CD8+ T-lymphocytes. Is the efficiency enough for clinical use?
2. Please briefly discuss about the life span of HA-1-specific CD8+ T-lymphocytes after in vivo infusion.
3. In the discussion section, the authors made mention of antibody against murine TCR constant chains. How about cytotoxicity by unmanipulated CD8+ cells because of murinization of transgenic TCR constant chains? Please discuss this possibility.
Minor comments:
1. line 121-122: please correct the duplication of GM-CSF.
2. line 131 and line 408: “described in [36].” , “published in [53, 54] and” Please avoid way for citation in the text.
3. Line 153: frameworks. [39, 40].→frameworks [39, 40].
4. Description of cell number; for example, line 132: 2*105 to 1*106 per well. Please correct as 2×105 to 1×106 cells per well in all cases.
Reviewer 2 Report
Opening sentences of the abstracts "A relapse of the malignant disease occurs in about half of the cases after allogeneic hematopoietic stem cell transplantation" and "More than half of allogeneic hematopoietic stem cell transplantations (allo-HSCT) result in relapse of malignant disease" are inaccurate and misleading. A significant proportion of HSCT patients relapse, but not one half.
Again, in the introduction, the sentence "However, approximately half of the patients experience disease relapse" is inaccurate.
That said, the whole introduction should be revised, as "anti relapse therapies" include an arsenal much larger than the antibodies described in the text, and the sentence "Apart from allo-HSCT, the only approved therapy for AML is 49 gemtuzumab ozogamicin" is grossly wrong. Please get on touch with AML clinical experts to re-write this section.
In the results, please describe in more detail differences in sensitivies among myeloid vs lymphoid diseases, and provide data for every single sample. The sentence "While the majority of PBMC samples belonged to patients with AML (supplementary table 4), the cytotoxic effect was not dependent on the diagnosis and/or blast percentage" is not enough. I cannot see suppl. Table 4, and suppl. Fig. 11 is not of help.
In the discussion, please consider that the current major limitation of CAR-T cell therapy is in solid tumors. Are TCR-T cells potentially more active?
Round 2
Reviewer 2 Report
The manuscript has been significantlly improved. In the web abstract, though, I still read the wrong sentence "" that must be corrected.